# Influence of Green Roofs on the Design of a Public Stormwater Drainage System: A Case Study

Flora Silva [1,2,3,*], Cristina Sousa Coutinho Calheiros [4], Guilherme Valle [1,5], Pedro Pinto [6], António Albuquerque [2,3,7] and Ana Maria Antão-Geraldes [8,9,*]

1  ESTiG, Instituto Politécnico de Bragança, Campus de Santa Apolónia, 5300-253 Bragança, Portugal
2  FibEnTech, 6201-001 Covilhã, Portugal
3  GeoBioTec-UBI, 6201-001 Covilhã, Portugal
4  Interdisciplinary Centre of Marine and Environmental Research (CIIMAR/CIMAR), University of Porto, Novo Edifício do Terminal de Cruzeiros do Porto de Leixões, Avenida General Norton de Matos, S/N, 4450-208 Matosinhos, Portugal
5  Universidade Tecnológica Federal do Paraná, Campus Curitiba, Curitiba 80230-000, PR, Brazil
6  Câmara Municipal de Bragança, Forte S. João de Deus, 5300-263 Bragança, Portugal
7  Department of Civil Engineering and Architecture, University of Beira Interior, 6201-001 Covilhã, Portugal
8  Centro de Investigação de Montanha (CIMO), Instituto Politécnico de Bragança, Campus de Santa Apolónia, 5300-253 Bragança, Portugal
9  Laboratório Associado para a Sustentabilidade e Tecnologia em Regiões de Montanha (SusTEC), Instituto Politécnico de Bragança, Campus de Santa Apolónia, 5300-253 Bragança, Portugal
*  Correspondence: flora@ipb.pt (F.S.); geraldes@ipb.pt (A.M.A.-G.)

**Abstract:** In the face of excessive soil sealing and the occurrence of heavy rainfall in short time periods leading to flooding, it is becoming increasingly urgent to implement public resilient stormwater drainage systems. Green roofs have several advantages at different levels, of which this paper highlights the ability to retain rainwater, to reduce problems with flooding in peaks of rainfall, and to increase in urban green infrastructure with all the benefits associated. In this sense, green roofs' impact on the design of a public stormwater drainage system and their implications for urban stormwater management was analyzed when compared with conventional roofs. If green roofs are used on the buildings in the study urban area, which has about 2.1 ha and is located in rainfall region B of Portugal, then the weighted average runoff coefficient ($C_m$) for the study area is 0.59. This scenario leads to a reduction in the maximum flow rate of 15.89% compared to the use of conventional roofs, with a $C_m$ of 0.70 for the same area. Thus, the use of green roofs instead of conventional roofs can have positive impacts on the surface runoff in urban areas and contribute to more sustainable urban drainage.

**Keywords:** sustainable urban drainage; public stormwater drainage system; green roofs; runoff coefficient





## 1. Introduction

Climate change has triggered an increase in the frequency and intensity of extreme weather events across several regions of the world (e.g., heat waves, droughts, high rainfall in short time intervals, extratropical storms and cyclones, with exceptionally intense winds that trigger various types of disasters, such as the destruction of housing and infrastructure, floods, mudslides, and landslides) [1–3]. In addition, the urban areas and drainage systems have been significantly altered by the increase in impermeable surfaces, building rooftops, and the disappearance of blue and green spaces, which simultaneously reduce urban resilience and anti-flooding capacity [4].

Stormwater management is usually carried out in a linear process and aims to discharge stormwater quickly to avoid flooding. However, when rainfall exceeds the capacity of drainage systems, widespread flooding eventually occurs in urban spaces. On the other

hand, when dry weather persists, there is a need to water the existing green spaces in order to maintain the vegetation, and water is used again in a linear fashion [5]. The difficulty of circular water management in urban areas makes these spaces particularly vulnerable to extreme droughts and floods. It is therefore urgent to increase the resilience of cities by promoting measures to make them "Water Wise Cities" [6].

The implementation of green infrastructure as a complement to grey infrastructure is crucial to achieve this goal. These green infrastructures include nature-based solutions (NBS) that ensure various environmental and socio-economic services. Green roofs (GR) are NBS that can be used in conjunction with other tools aimed at promoting the circular economy of water in urban spaces [7,8]. They can be constructed at ground level or on top of buildings, following technical and scientific guidelines. They result from planting vegetation on a substrate followed by several layers of other materials on that site on a built structure. Based on the substrate depth, GR can be classified as extensive (<15 cm), semi-intensive (>15 and <25 cm), or intensive (>25 cm) [7,9,10]. Extensive GR are implemented more frequently because they are lighter, cheaper, and require less maintenance than intensive GR [11,12].

GR are efficient solutions for flood mitigation as they delay the peak flow of stormwater by releasing the water gradually (sponge effect) and avoid overloading the stormwater drainage system. Part of this water infiltrates and is retained in the substrates and drainage layer, being released during dry periods by evapotranspiration [5,9]. A panoply of experimental studies have demonstrated, in several regions of the world and in different climates, the hydrological performance of GR in terms of control of rainwater runoff, sometimes in comparison with conventional roofs [13–30]. Noteworthy are two studies carried out in Lisbon, Portugal, under the Mediterranean climate, where the hydrological performance of GR was analyzed with favorable results in terms of runoff control, highlighting among other parameters, water retention from 37 to 100% in the study of Santos et al. [31] and from 12 to 100% in the study of Brandão et al. [32].

Moreover, besides being important tools for circular urban water management, GR also offer potential benefits in terms of aesthetic value, biodiversity conservation, noise buffering, air pollution mitigation, and "heat island" effect reduction, promoting energy efficiency and reduction of $CO_2$, and other greenhouse gas emissions [7,19,25,27,33].

Indeed, according to a very recent study by Bona et al. [34], in Europe continuous efforts are being made to increase and integrate NBS for mitigation and adaptation to climate challenges, with the development of resilience in cities aiming at sustainability. Its implementation is directly aligned with the goals of the United Nations 2030 Agenda, playing an active role in the strategic implementation and achievement of the Sustainable Development Goals (SDGs) [35].

The integration of the GR concept in the design of public stormwater drainage systems has not yet been deeply exploited. There is still a lack of knowledge by the public and private sectors and the general public about the high potential of the contribution of this natural engineering tool for water management in urban spaces, as well as the technical aspects and socio-economic benefits. Accordingly, the novelty of this study is to analyze the influence that the use of extensive GR in relation to the use of conventional roofs has on the hydraulic design of a public stormwater drainage network in an urban allotment. Concomitantly, a cost estimate of the public stormwater drainage network designed for three scenarios will be presented.

## 2. Materials and Methods

### 2.1. Study Area

The urban allotment is located in Macedo de Cavaleiros (Latitude: 41°32'11'' N; Longitude: 6°57'22'' W; altitude: 572 m), a city in the Northeast of Portugal, located about 40 km from the district capital which is Bragança. The municipality has 14,251 inhabitants [36]. The climate is continental with Mediterranean influences. Annual mean precipitation is around 700 mm per year, occurring mainly in autumn and winter but in a very irregular

pattern [37]. It has an area of 21,038.37 m², consisting of streets (roads, car parks, and pavements), green areas, and plots (buildings and building's backyard) (Figure 1).

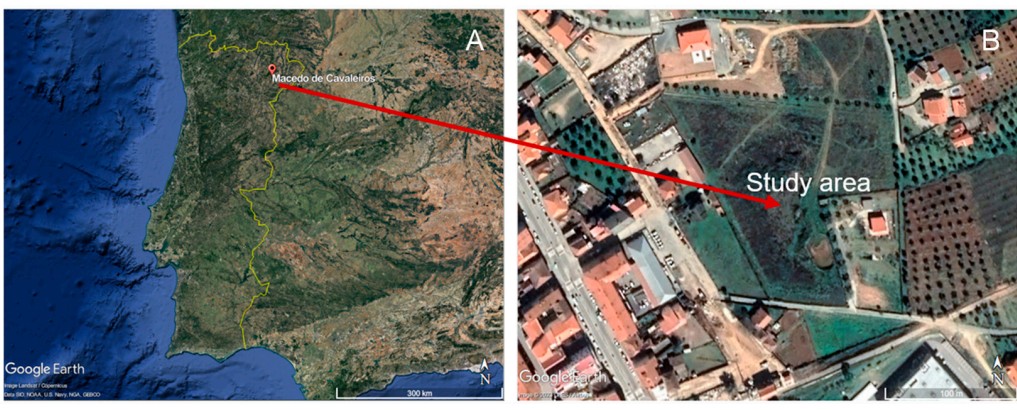

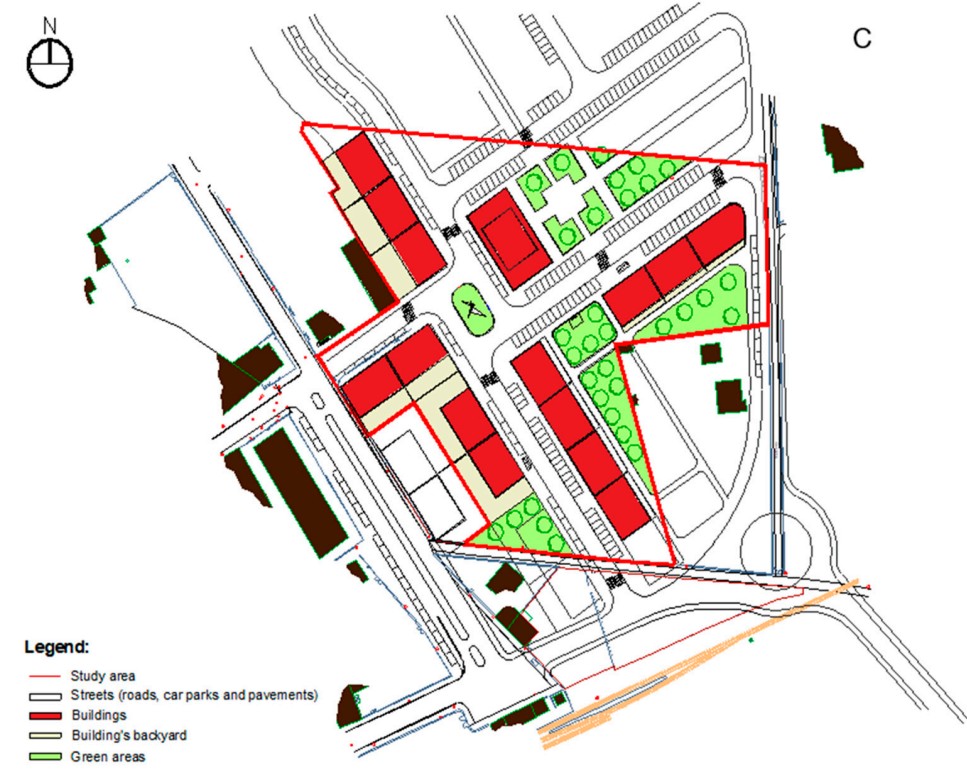

**Figure 1.** Location of Macedo de Cavaleiros, Portugal (**A**) and the study area (**B**); urban allotment (not to scale) (**C**).

## 2.2. Criteria for Sizing the Public Stormwater Drainage System

The public stormwater drainage network, of the separative type, will be sized in accordance with the Portuguese Regulatory Decree no. 23/95 of 23 August [38] and adequate guidelines [39]. It will comprise, among others, individual connection pipes, storm drains, manholes, and drainage pipes. The drainage pipes, represented by lines, will be designated DPi, and the manholes, represented by circles, will be designated Di. The minimal drainage pipe covering will be 1.00 m and the dimensioning will be carried out without exceeding 4.00 m in depth.

Sizing begins with the determination of the effective area, the concentration time, and the rainfall intensity.

The effective area is the product of the drained area and the runoff coefficient. The drained area will be measured in the drawing using AutoCAD® software. The runoff coefficients adopted for each surface are shown in Table 1 and were defined according to [39–41]. It should be noted that in impermeable roofs, where the water loss by absorption is reduced and where the water retained or evaporated is not significant, the runoff coefficient has a value close to 1.0. In the case of GR, average annual values between 0.4 and 0.6 for extensive GR are generally adopted in central Europe and the United Kingdom [41].

Three scenarios for the sizing of the public stormwater drainage network were considered:

- Scenario 1—buildings with sloped, impermeable, and smooth roofs (e.g., ceramic tile, Figure 2A,D);
- Scenario 2—buildings with flat roofs with protective aggregates (e.g., gravel, Figure 2B,E);
- Scenario 3—buildings with extensive GR, without irrigation (substrate thickness ≤ 15 cm, Figure 2C,F).

**Table 1.** Area of allotment surfaces and runoff coefficients.

| Surface | | Area (m²) | Runoff Coefficient (Dimensionless) |
|---|---|---|---|
| Streets | Roads<br>Car parks<br>Pavements | 9928.65 | 0.80 |
| Green areas | | 3428.55 | 0.20 |
| Plots | Buildings | 5769.85 | 0.90 [1]; 0.70 [2]; 0.50 [3] |
| | Building's backyard (50% paved area) | 955.66 | 0.80 |
| | Building's backyard (50% green area) | 955.66 | 0.20 |

Note: [1] Scenario 1: sloped, impermeable, and smooth roofs; [2] Scenario 2: flat roofs with protective aggregates; [3] Scenario 3: extensive green roofs.

However, given the fact that the area considered is made up of surfaces of different characteristics, a weighted average runoff coefficient was determined, according to Equation (1) [39].

$$C_m = \frac{\sum_{i=1}^{N} S_i \times C_i}{\sum_{i=1}^{N} S_i} \tag{1}$$

where $S_i$ is the surface of area $i$ (m²), $C_i$ is the runoff coefficient of the area $i$ (dimensionless), and $C_m$ is the weighted average runoff coefficient (dimensionless).

The concentration time results from adding the entry time to the path time. The drainage basin is considered sloping (1.5% to 8%, with impermeable areas greater than 50%), so the time considered for the entry of water into the drainage pipes will be 7.5 min [39].

For rainfall intensity, considering a return period of 10 years, the intensity–duration–frequency (IDF) curve is adopted ($I = 232.21 \times t^{-0.549}$, rainfall region B, Portugal) and using the calculated concentration times [38,39].

Using the values of the effective areas and the rainfall intensities, the maximum design flow rates are calculated by the rational method (Equation (2)) [39], and the minimum design flow rates, corresponding to 30% of the maximum design flow rates and with a minimum of 5 L/s.

$$Q = C \times I \times A \tag{2}$$

where $Q$ is the design flow rate (L/s), $I$ is the average rainfall intensity (mm/h), $C$ is the runoff coefficient (dimensionless), and $A$ is the drained area (m²).

In flow with the free surface, the most common equation to analyze the hydraulic performance is the Manning-Strickler equation (Equation (3)) [39]:

$$Q = A_m \times K_s \times R_h^{\frac{2}{3}} \times i^{\frac{1}{2}} \tag{3}$$

where $Q$ is the flow rate (m$^3$/s), $A_m$ is the flow section area or wetted area (m$^2$), $K_s$ is the roughness coefficient, using the value of 110 m$^{\frac{1}{3}}$/s [39], $R_h$ is the hydraulic radius (m), and $i$ is the drainage pipe inclination (m/m). As the drainage pipes will be in corrugated polypropylene, the diameters are shown in Table 2 [42].

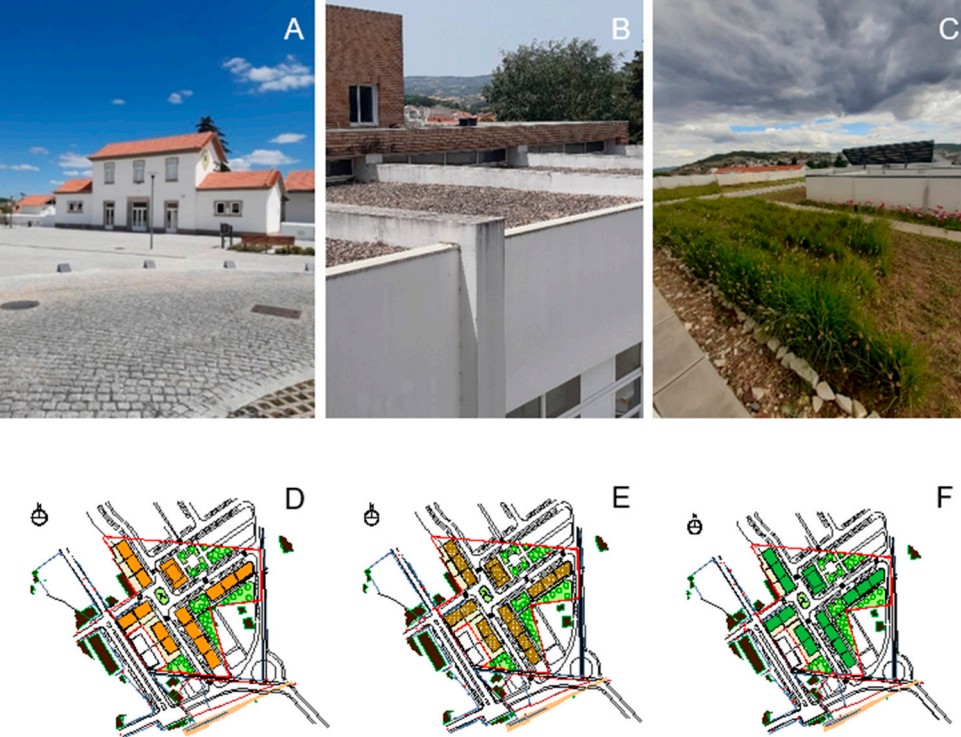

**Figure 2.** Examples of buildings with sloped, impermeable, and smooth roofs (**A**); flat roofs with protective aggregates (**B**); and extensive green roofs (**C**) in the cities of Macedo de Cavaleiros and Bragança, Northeast Portugal. Layout of the urban allotment for the Scenario 1 (**D**); Scenario 2 (**E**), and Scenario 3 (**F**).

**Table 2.** Diameters of stormwater drainage pipes.

| Drainage Pipe | Nominal Diameter (mm) | Inside Diameter (mm) | Wall Thickness (mm) |
|---|---|---|---|
| | 200 | 177 | 11.5 |
| | 250 | 224 | 13.0 |
| | 315 | 275 | 20.0 |
| | 400 | 352 | 24.0 |
| | 500 | 443 | 28.5 |

The diameters, velocities, inclinations, runoff heights (liquid blade), drag tensions, and path times are then calculated.

The Portuguese Regulatory Decree no. 23/95 of 23 August [38] requires a minimum nominal diameter for drainage pipes of 200 mm. Considering the design flow rates, the same decree imposes a minimum flow velocity of 0.9 m/s and a maximum of 5.0 m/s. The inclination of the drainage pipes should not be, in general, less than 0.3% and no more than 15.0%.

It is considered that the height of the liquid blade can be equal to the diameter of the drainage pipe. For the drag tension, a minimum value of 4 N/m$^2$ is considered [38,39].

In hydraulic sizing, cost minimization must be considered, achieved through a careful combination of diameters, while respecting regulatory impositions [38]. The stipulated

prices for the various works are those practiced in the region in January 2023, and prices were also requested for the various corrugated polypropylene drainage pipe diameters [42].

All calculations were performed using MS Excel.

## 3. Results and Discussion

### 3.1. Sizing the Public Stormwater Drainage System

Figure 3 presents, in blue color, the layout of the designed public stormwater drainage network, considering the layout of scenario 3. The following weighted average runoff coefficients were obtained: $C_m = 0.70$, buildings with sloped, impermeable, and smooth roofs (scenario 1), $C_m = 0.65$, buildings with flat roofs with protective aggregates (scenario 2), and $C_m = 0.59$, buildings with extensive GR and without irrigation (scenario 3).

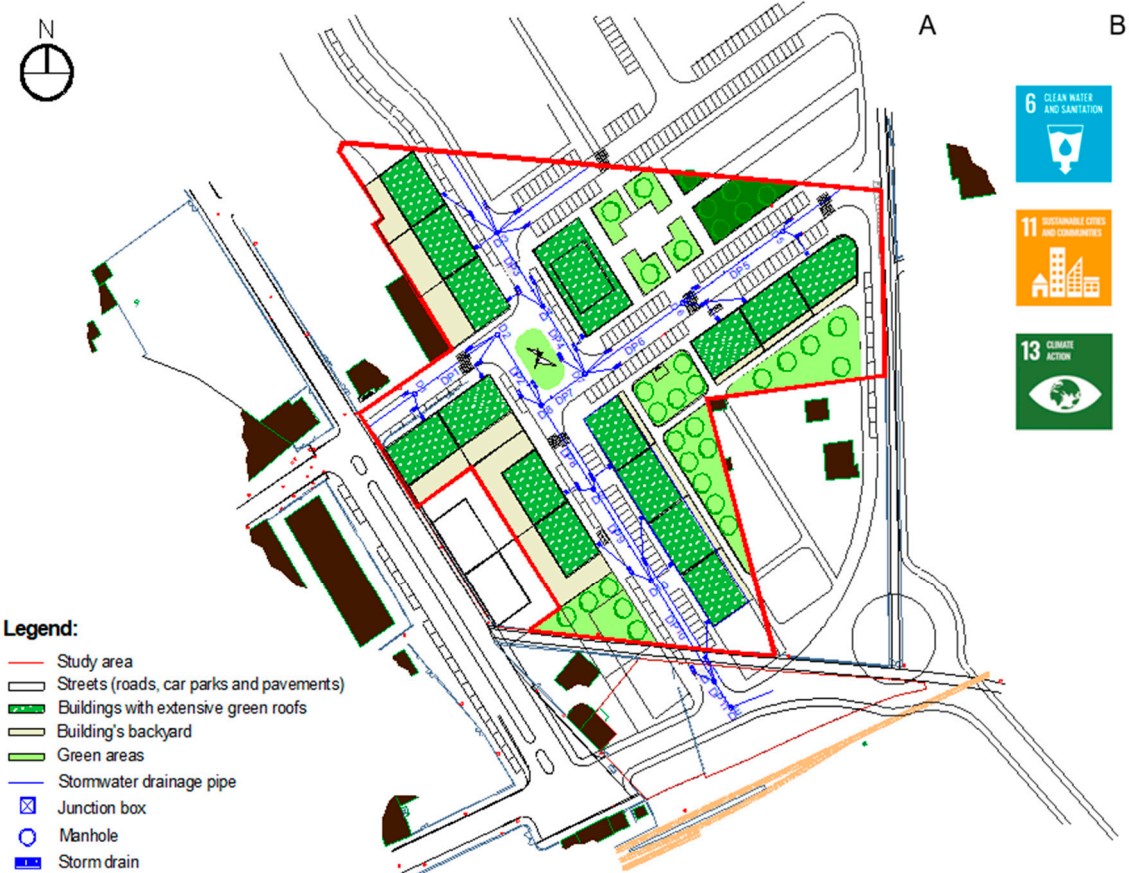

**Figure 3.** Stormwater drainage network (not to scale) (**A**); sustainable development goals addresses in the case study (**B**) [35].

The values of the effective area, concentration time, and rainfall intensity are shown in Table 3. The area drained by the DP11 drainage pipe is 24,038.37 m$^2$ because the manholes D1, D3, and D5 receive flows from areas with 1000 m$^2$ each.

As expected, the effective areas decrease according to the application of a lower weighted average flow coefficient (0.70, 0.65, and 0.59 for Scenarios 1, 2, and 3, respectively). However, a limitation of this study is the inexistence of a value of the runoff coefficient for the Northeast region of Portugal applied to GR, which was stipulated in the literature [40,41]. This information can be obtained by performing experimental tests in this region using a prototype GR. The studies of [41,43] present a formula that can be useful for the in situ determining the monthly runoff coefficient as a function of different variables. It is also possible to verify an increase in the concentration time for Scenario 3

because a GR has a direct effect on stormwater runoff when compared with a conventional roof in a way that lowers (attenuates) and delays peak runoff [7,15,25–27].

**Table 3.** Effective area, concentration time, and rainfall intensity for the 3 scenarios.

| Drainage Pipe | Drained Area (m$^2$) | Effective Area (m$^2$) | | | Concentration Time (min) | | | Rainfall Intensity (mm/h) | | |
|---|---|---|---|---|---|---|---|---|---|---|
| Scenarios: | | (1) | (2) | (3) | (1) | (2) | (3) | (1) | (2) | (3) |
| DP1 | 1712.31 | 1202.71 | 1108.79 | 1014.87 | 7.50 | 7.50 | 7.50 | 76.82 | 76.82 | 76.82 |
| DP2 | 2579.77 | 1812.00 | 1670.50 | 1529.00 | 7.80 | 7.81 | 7.81 | 75.18 | 75.14 | 75.11 |
| DP3 | 3789.24 | 2661.53 | 2453.68 | 2245.84 | 7.50 | 7.50 | 7.50 | 76.82 | 76.82 | 76.82 |
| DP4 | 4914.52 | 3451.91 | 3182.35 | 2912.78 | 7.62 | 7.63 | 7.63 | 76.13 | 76.12 | 76.10 |
| DP5 | 1854.01 | 1302.24 | 1200.54 | 1098.85 | 7.50 | 7.50 | 7.50 | 76.82 | 76.82 | 76.82 |
| DP6 | 4350.22 | 3055.55 | 2816.94 | 2578.33 | 7.93 | 7.96 | 7.99 | 74.48 | 74.36 | 74.21 |
| DP7 | 12,069.89 | 8477.78 | 7815.74 | 7153.70 | 7.73 | 7.75 | 7.78 | 75.53 | 75.43 | 75.30 |
| DP8 | 15,135.70 | 10,631.18 | 9800.97 | 8970.77 | 8.02 | 8.02 | 8.06 | 74.02 | 74.02 | 73.85 |
| DP9 | 17,340.43 | 12,179.76 | 11,228.62 | 10,277.49 | 8.15 | 8.16 | 8.21 | 73.37 | 73.32 | 73.09 |
| DP10 | 19,396.35 | 13,623.82 | 12,559.91 | 11,496.01 | 8.28 | 8.30 | 8.36 | 72.77 | 72.67 | 72.38 |
| DP11 | 24,038.37 (*) | 16,884.33 | 15,565.81 | 14,247.28 | 8.45 | 8.51 | 8.50 | 71.94 | 71.66 | 71.71 |

Note: (1) Scenario 1; (2) Scenario 2; (3) Scenario 3. (*) The drained area of the DP11 drainage pipe is 24,038.37 m$^2$ because the manholes D1, D3, and D5 are supplied by areas of 1000 m$^2$ each.

The results obtained for the hydraulic calculation and the elements for the implementation of the drainage pipes and manholes are presented in Appendix A (Tables A1 and A2 (Scenario 1), Tables A3 and A4 (Scenario 2), and Tables A5 and A6 (Scenario 3)).

Tables A1, A3 and A5 show that the minimum self-cleaning flow rate of 5.0 L/s was ensured in all scenarios. The maximum flow rates obtained in the last drainage pipe were 337.43, 309.84, and 283.81 L/s for Scenarios 1, 2, and 3, respectively. As expected, the flow rates show reductions, since the fact that the effective area decreases according to the type of roof considered in each scenario. Thus, a flow rate reduction of 15.89% is observed if extensive GR are used instead of sloped, impermeable, and smooth roofs (Scenario 1), and of 8.40% if extensive GR are used instead of flat roofs with protective aggregates (Scenario 2). It is a fact that conventional roofs allow rainwater to runoff their surfaces quickly, increasing flooding. GR, on the other hand, are considered to be of greater importance as they absorb rainwater, delaying its runoff and promoting evapotranspiration, increasing the effectiveness of stormwater management [12], as it is also shown in the studies [22,23,31,32]. Despite the urban allotment of only 2.1 ha, with the reduction of the maximum flow rate obtained, the impact of this NBS at the city level could be huge. Indeed, according to [32], if 75% of the flat roof area of the municipality of Lisbon were greened, approximately 166,500–224,000 m$^3$ of water could be retained during extreme precipitation events, relieving the drainage systems and preventing floods. Similar results were reported by [44] for the city of London.

As observed in Tables A1, A3 and A5, the minimum (0.90 m/s) and maximum (5.00 m/s) velocities are verified, as well as the minimum (0.3%) and maximum (15.0%) inclinations. The drag tensions also complied with the criterion of 4 N/m$^2$ minimum.

For the three scenarios, up to drainage pipe CP6, the diameter is always the same (Ø 200 mm). There is a variation of diameters in the drainage pipes CP7, CP10, and CP11, with a decrease in diameter for the case of GR. The maximum diameter value is 500 mm (Scenario 1), 500 mm (Scenario 2), and 400 mm (Scenario 3), in the last drainage pipe. In Xu et al. [4] it is mentioned that the solution to flooding problems has predominantly been to increase the drainage pipes diameters to drain stormwater as quickly as possible, which may not be the most effective solution. Thus, to solve this problem, and as seen in Scenario 3, it is plausible to consider the integration of an NBS into public storm drainage systems.

Using GR, and by observing Tables A2, A4 and A6, the drainage pipes' diameters and manhole depths were smaller in some sections, hence the land moving works value also decreased when compared to the use of conventional roofs.

*3.2. Cost Analysis of the Public Stormwater Drainage System*

Cost estimation for the designed public stormwater drainage network is presented in Table 4. The costs for work 1—"land moving", refer to (1) excavation for opening trenches for laying the pipe; (2) regularization of the bottom of trenches; (3) backfilling with borrow land involving the pipe; (4) backfilling trenches with land from the excavation of the trenches itself; and (5) loading, transporting, and unloading of surplus products to an authorized dump site [45]. In work 2—"pipes", the cost of supplying and laying the piping is considered. Concerning work 3—"network components", the cost of constructing manholes, storm drains, and a connection to the existing network was estimated, as well as the execution of individual connection pipes for stormwater drainage. In work 4—"other works", the cost of supplying and laying pre-signaling tape and carrying out final tests on the stormwater drainage system was estimated.

**Table 4.** Budget summary for the 3 scenarios.

| Work Description | Partial Price (EUR) | | |
|---|---|---|---|
| | Scenario 1 | Scenario 2 | Scenario 3 |
| 1. Land moving | 6546.20 | 6788.41 | 5788.87 |
| 2. Pipes | 9078.00 | 8983.50 | 8364.50 |
| 3. Network components | 14,500.00 | 14,500.00 | 14,500.00 |
| 4.1. Other works | 847.20 | 847.20 | 847.20 |
| Partial price (EUR) | 30,971.40 | 31,119.11 | 29,500.57 |
| Total value (EUR) | 30,971.40 + VAT | 31,119.11 + VAT | 29,500.57 + VAT |

Therefore, the cost for the implementation of the public stormwater drainage system is slightly lower for Scenario 3, with extensive GR (EUR 29,500.57 + VAT) when compared to Scenario 1, with sloped, impermeable, and smooth roofs (EUR 30,971.40 + VAT) and Scenario 2, with flat roofs with protective aggregates (EUR 31,119.11 + VAT).

Given the results obtained and the difference in costs between the scenarios being irrelevant, the installation of GR in buildings can have positive effects on urban stormwater management, since a reduction of 15.89% in the maximum flow rate is expected. However, as in the present study, the hydraulic sizing of the network was carried out based on some theoretical parameters, such as the runoff coefficient, as there is a need to develop a GR prototype for the region under study in order to collect experimental data to obtain this coefficient, depending on the characteristics of the GR, air temperature, and precipitation [41]. Such data will make it possible to gain knowledge about the hydrological performance of GR, validate and extrapolate the data from the present study, whether hydraulic or cost, and encourage policy makers and other stakeholders to define guidelines for the integration of these NBS upstream of new public stormwater drainage systems. Promoting its implementation could help, among other aspects [7,9], to mitigate urban flooding, which has been occurring more frequently, and to reduce the associated economic, environmental, and social losses.

## 4. Conclusions

Given the current context of climate change, it is crucial that stormwater management responds to the various challenges and pressures to which water resources and cities are exposed. It is, therefore, necessary to reduce impermeable urban areas, promoting temporary water retention and minimizing the impacts of urban development on the natural environment. Best practices for sustainable urban management include resilient structures such as GR. In addition to mitigating flood peaks in urban areas, and reducing the consequences of exceptional rainfall events, these NBS increase urban green infrastructure, with other associated benefits.

This work allowed us to conclude that impermeable areas have a relevant influence in the study area, located in Northeast Portugal. Using GR in an urban allotment with

2.1 ha would result in a weighted average runoff coefficient of 0.59, which would reduce stormwater flow by around 15.9% in relation to the use of conventional roofs. It may also contribute as a sustainable urban drainage technique to be explored in the future for other urban areas.

**Author Contributions:** Conceptualization, F.S. and A.M.A.-G.; methodology, F.S. and A.M.A.-G.; software, G.V. and F.S.; formal analysis, F.S., G.V. and A.M.A.-G.; investigation, G.V., P.P., F.S. and A.M.A.-G.; resources, F.S. and P.P.; data curation, F.S., G.V. and A.M.A.-G.; writing—original draft preparation, F.S. and A.M.A.-G.; writing—review and editing, F.S., A.M.A.-G., A.A. and C.S.C.C.; visualization F.S., A.M.A.-G., A.A. and C.S.C.C.; supervision, F.S. and A.M.A.-G.; project administration, F.S. and A.M.A.-G. All authors have read and agreed to the published version of the manuscript.

**Funding:** This research was funded by the Foundation for Science and Technology (FCT, Portugal) for financial support by national funds FCT/MCTES (PIDDAC) to CIMO (UIDB/00690/2020 and UIDP/00690/2020), SusTEC (LA/P/0007/2020), FibEnTech (UIDB/00195/2020), GeoBioTec (UIDB/04035/2020) and CIIMAR (UIDB/04423/2020 and UIDP/04423/2020).

**Institutional Review Board Statement:** Not applicable.

**Informed Consent Statement:** Not applicable.

**Data Availability Statement:** Not applicable.

**Acknowledgments:** The materials for the case study provided by Dolmu—Arquitetura | Engenharia, in the person of the Architect Paulo Moreira, were appreciated.

**Conflicts of Interest:** The authors declare no conflict of interest.

## Appendix A

**Table A1.** Hydraulic calculation for Scenario 1.

| Drainage Pipe | Flow Rate (L/s) | | Inside Diameter (mm) | Inclination (%) | Velocity (m/s) | | Drain Height (m) | | Drag Tension (N/m²) | | Path Time (min) | |
|---|---|---|---|---|---|---|---|---|---|---|---|---|
| | Min. | Max. | | | Min. | Max. | Min. | Max. | Min. | Max. | Min. | Max. |
| DP1 | 7.70 | 25.66 | 177.00 | 1.99 | 1.43 | 1.98 | 0.048 | 0.092 | 5.43 | 8.86 | 0.30 | 0.42 |
| DP2 | 11.35 | 37.84 | 177.00 | 6.34 | 2.41 | 3.36 | 0.044 | 0.083 | 15.92 | 26.27 | 0.14 | 0.20 |
| DP3 | 17.04 | 56.79 | 177.00 | 7.53 | 2.88 | 3.96 | 0.051 | 0.100 | 21.72 | 35.14 | 0.12 | 0.17 |
| DP4 | 21.90 | 73.00 | 177.00 | 6.18 | 2.87 | 3.85 | 0.062 | 0.127 | 20.64 | 32.00 | 0.12 | 0.16 |
| DP5 | 8.34 | 27.79 | 177.00 | 9.51 | 2.54 | 3.59 | 0.034 | 0.062 | 19.10 | 32.08 | 0.20 | 0.28 |
| DP6 | 18.97 | 63.22 | 177.00 | 3.49 | 2.25 | 2.93 | 0.066 | 0.145 | 12.35 | 18.40 | 0.24 | 0.32 |
| DP7 | 53.36 | 177.87 | 275.00 | 2.63 | 2.62 | 3.41 | 0.103 | 0.225 | 14.49 | 21.58 | 0.09 | 0.12 |
| DP8 | 65.58 | 218.59 | 275.00 | 3.97 | 3.22 | 4.20 | 0.103 | 0.225 | 21.88 | 32.59 | 0.13 | 0.17 |
| DP9 | 74.47 | 248.24 | 275.00 | 5.13 | 3.65 | 4.76 | 0.103 | 0.225 | 28.23 | 42.03 | 0.13 | 0.17 |
| DP10 | 82.62 | 275.39 | 352.00 | 3.49 | 3.21 | 4.40 | 0.109 | 0.216 | 21.17 | 33.85 | 0.15 | 0.21 |
| DP11 | 101.23 | 337.43 | 443.00 | 2.24 | 2.85 | 3.40 | 0.125 | 0.272 | 15.78 | 27.34 | 0.05 | 0.06 |

**Table A2.** Elements for deploying drainage pipes for Scenario 1.

| Drainage Pipe | Manhole | | Nominal Diameter (mm) | Length (m) | Terrain Elevation (m) | | Manhole Base (m) | | Manhole Depth (m) | | Drainage Pipe Covering (m) | |
|---|---|---|---|---|---|---|---|---|---|---|---|---|
| | (1) | (2) | | | (1) | (2) | (1) | (2) | (1) | (2) | (1) | (2) |
| DP1 | D1 | D2 | 200.00 | 35.70 | 47.51 | 46.80 | 46.32 | 45.61 | 1.19 | 1.19 | 1.00 | 1.00 |
| DP2 | D2 | D8 | 200.00 | 28.40 | 46.80 | 45.00 | 45.61 | 42.73 | 1.19 | 2.27 | 1.00 | 1.00 |
| DP3 | D3 | D4 | 200.00 | 29.60 | 49.03 | 46.80 | 47.84 | 45.61 | 1.19 | 1.19 | 1.00 | 1.00 |
| DP4 | D4 | D7 | 200.00 | 27.20 | 46.80 | 45.12 | 45.61 | 43.28 | 1.19 | 1.84 | 1.00 | 1.00 |
| DP5 | D5 | D6 | 200.00 | 43.00 | 50.09 | 46.00 | 48.90 | 44.81 | 1.19 | 1.19 | 1.00 | 1.00 |
| DP6 | D6 | D7 | 200.00 | 42.50 | 46.00 | 45.12 | 44.81 | 43.28 | 1.19 | 1.84 | 1.00 | 1.60 |
| DP7 | D7 | D8 | 315.00 | 18.90 | 45.12 | 45.00 | 43.28 | 42.73 | 1.84 | 2.27 | 1.60 | 1.98 |
| DP8 | D8 | D9 | 315.00 | 33.70 | 45.00 | 43.80 | 42.73 | 41.39 | 2.27 | 2.41 | 1.98 | 2.12 |
| DP9 | D9 | D10 | 315.00 | 37.00 | 43.80 | 42.80 | 41.39 | 39.49 | 2.41 | 3.31 | 2.12 | 3.01 |
| DP10 | D10 | D11 | 400.00 | 40.50 | 42.80 | 39.37 | 39.49 | 37.99 | 3.31 | 1.38 | 3.01 | 1.00 |
| DP11 | D11 | DE [*] | 500.00 | 10.70 | 39.37 | 39.13 | 37.99 | 37.75 | 1.38 | 1.38 | 1.00 | 1.00 |

Note: (1) upstream; (2) downstream; [*] existing manhole.

**Table A3.** Hydraulic calculation for Scenario 2.

| Drainage Pipe | Flow Rate (L/s) | | Inside Diameter (mm) | Inclination (%) | Velocity (m/s) | | Drain Height (m) | | Drag Tension (N/m$^2$) | | Path Time (min) | |
|---|---|---|---|---|---|---|---|---|---|---|---|---|
| | Min. | Max. | | | Min. | Max. | Min. | Max. | Min. | Max. | Min. | Max. |
| DP1 | 7.10 | 23.66 | 177.00 | 1.99 | 1.39 | 1.94 | 0.046 | 0.088 | 5.24 | 8.60 | 0.31 | 0.43 |
| DP2 | 10.46 | 34.87 | 177.00 | 6.34 | 2.35 | 3.29 | 0.042 | 0.079 | 15.37 | 25.44 | 0.14 | 0.20 |
| DP3 | 15.71 | 52.36 | 177.00 | 7.53 | 2.81 | 3.89 | 0.049 | 0.095 | 20.98 | 34.14 | 0.13 | 0.18 |
| DP4 | 20.19 | 67.29 | 177.00 | 6.18 | 2.81 | 3.80 | 0.059 | 0.120 | 19.95 | 31.36 | 0.12 | 0.16 |
| DP5 | 7.69 | 25.62 | 177.00 | 9.51 | 2.48 | 3.51 | 0.032 | 0.060 | 18.43 | 31.01 | 0.20 | 0.29 |
| DP6 | 17.45 | 58.18 | 177.00 | 2.95 | 2.07 | 2.70 | 0.066 | 0.145 | 10.46 | 15.58 | 0.26 | 0.34 |
| DP7 | 49.13 | 163.75 | 224.00 | 6.66 | 3.63 | 4.74 | 0.084 | 0.184 | 29.87 | 44.49 | 0.07 | 0.09 |
| DP8 | 60.46 | 201.53 | 275.00 | 3.38 | 2.97 | 3.87 | 0.103 | 0.225 | 18.60 | 27.70 | 0.15 | 0.19 |
| DP9 | 68.61 | 228.71 | 275.00 | 4.35 | 3.37 | 4.39 | 0.103 | 0.225 | 23.96 | 35.68 | 0.14 | 0.18 |
| DP10 | 76.06 | 253.54 | 352.00 | 3.38 | 3.10 | 4.26 | 0.105 | 0.207 | 19.90 | 32.02 | 0.16 | 0.22 |
| DP11 | 92.95 | 309.84 | 443.00 | 2.24 | 2.78 | 3.29 | 0.119 | 0.261 | 15.22 | 26.76 | 0.05 | 0.06 |

**Table A4.** Elements for deploying drainage pipes for Scenario 2.

| Drainage Pipe | Manhole | | Nominal Diameter (mm) | Length (m) | Terrain Elevation (m) | | Manhole Base (m) | | Manhole Depth (m) | | Drainage Pipe Covering (m) | |
|---|---|---|---|---|---|---|---|---|---|---|---|---|
| | (1) | (2) | | | (1) | (2) | (1) | (2) | (1) | (2) | (1) | (2) |
| DP1 | D1 | D2 | 200.00 | 35.70 | 47.51 | 46.80 | 46.32 | 45.61 | 1.19 | 1.19 | 1.00 | 1.00 |
| DP2 | D2 | D8 | 200.00 | 28.40 | 46.80 | 45.00 | 45.61 | 42.19 | 1.19 | 2.81 | 1.00 | 1.00 |
| DP3 | D3 | D4 | 200.00 | 29.60 | 49.03 | 46.80 | 47.84 | 45.61 | 1.19 | 1.19 | 1.00 | 1.00 |
| DP4 | D4 | D7 | 200.00 | 27.20 | 46.80 | 45.12 | 45.61 | 43.51 | 1.19 | 1.61 | 1.00 | 1.00 |
| DP5 | D5 | D6 | 200.00 | 43.00 | 50.09 | 46.00 | 48.90 | 44.81 | 1.19 | 1.19 | 1.00 | 1.00 |
| DP6 | D6 | D7 | 200.00 | 42.50 | 46.00 | 45.12 | 44.81 | 43.51 | 1.19 | 1.61 | 1.00 | 1.37 |
| DP7 | D7 | D8 | 250.00 | 18.90 | 45.12 | 45.00 | 43.51 | 42.19 | 1.61 | 2.81 | 1.37 | 2.51 |
| DP8 | D8 | D9 | 315.00 | 33.70 | 45.00 | 43.80 | 42.19 | 41.05 | 2.81 | 2.75 | 2.51 | 2.45 |
| DP9 | D9 | D10 | 315.00 | 37.00 | 43.80 | 42.80 | 41.05 | 39.44 | 2.75 | 3.36 | 2.45 | 3.06 |
| DP10 | D10 | D11 | 400.00 | 40.50 | 42.80 | 39.37 | 39.44 | 37.99 | 3.36 | 1.38 | 3.06 | 1.00 |
| DP11 | D11 | DE [*] | 500.00 | 10.70 | 39.37 | 39.13 | 37.99 | 37.75 | 1.38 | 1.38 | 1.00 | 1.00 |

Note: (1) upstream; (2) downstream; [*] existing manhole.

**Table A5.** Hydraulic calculation for scenario 3.

| Drainage Pipe | Flow Rate (L/s) | | Inside Diameter (mm) | Inclination (%) | Velocity (m/s) | | Drain Height (m) | | Drag Tension (N/m$^2$) | | Path Time (min) | |
|---|---|---|---|---|---|---|---|---|---|---|---|---|
| | Min. | Max. | | | Min. | Max. | Min. | Max. | Min. | Max. | Min. | Max. |
| DP1 | 6.50 | 21.66 | 177.00 | 1.99 | 1.36 | 1.89 | 0.044 | 0.084 | 5.04 | 8.31 | 0.31 | 0.44 |
| DP2 | 9.57 | 31.90 | 177.00 | 6.34 | 2.29 | 3.21 | 0.040 | 0.075 | 14.78 | 24.56 | 0.15 | 0.21 |
| DP3 | 14.38 | 47.92 | 177.00 | 7.53 | 2.74 | 3.81 | 0.047 | 0.090 | 20.19 | 33.05 | 0.13 | 0.18 |
| DP4 | 18.47 | 61.57 | 177.00 | 6.18 | 2.74 | 3.73 | 0.056 | 0.112 | 19.22 | 30.56 | 0.12 | 0.17 |
| DP5 | 7.03 | 23.45 | 177.00 | 9.51 | 2.42 | 3.42 | 0.031 | 0.057 | 17.72 | 29.88 | 0.21 | 0.30 |
| DP6 | 15.94 | 53.15 | 177.00 | 2.46 | 1.89 | 2.46 | 0.066 | 0.145 | 8.73 | 13.00 | 0.29 | 0.38 |
| DP7 | 44.89 | 149.63 | 224.00 | 5.56 | 3.32 | 4.33 | 0.084 | 0.184 | 24.94 | 37.15 | 0.07 | 0.09 |
| DP8 | 55.21 | 184.02 | 275.00 | 2.82 | 2.71 | 3.53 | 0.103 | 0.225 | 15.51 | 23.10 | 0.16 | 0.21 |
| DP9 | 62.60 | 208.66 | 275.00 | 3.62 | 3.07 | 4.00 | 0.103 | 0.225 | 19.94 | 29.70 | 0.15 | 0.20 |
| DP10 | 69.34 | 231.15 | 275.00 | 5.54 | 3.68 | 4.91 | 0.097 | 0.203 | 29.14 | 44.87 | 0.14 | 0.18 |
| DP11 | 85.14 | 283.81 | 352.00 | 2.24 | 2.76 | 3.68 | 0.125 | 0.260 | 15.10 | 23.27 | 0.05 | 0.06 |

**Table A6.** Elements for deploying drainage pipes for Scenario 3.

| Drainage Pipe | Manhole | | Nominal Diameter (mm) | Length (m) | Terrain Elevation (m) | | Manhole Base (m) | | Manhole Depth (m) | | Drainage Pipe Covering (m) | |
|---|---|---|---|---|---|---|---|---|---|---|---|---|
| | (1) | (2) | | | (1) | (2) | (1) | (2) | (1) | (2) | (1) | (2) |
| DP1 | D1 | D2 | 200.00 | 35.70 | 47.51 | 46.80 | 46.32 | 45.61 | 1.19 | 1.19 | 1.00 | 1.00 |
| DP2 | D2 | D8 | 200.00 | 28.40 | 46.80 | 45.00 | 45.61 | 42.61 | 1.19 | 2.39 | 1.00 | 1.00 |
| DP3 | D3 | D4 | 200.00 | 29.60 | 49.03 | 46.80 | 47.84 | 45.61 | 1.19 | 1.19 | 1.00 | 1.00 |
| DP4 | D4 | D7 | 200.00 | 27.20 | 46.80 | 45.12 | 45.61 | 43.72 | 1.19 | 1.40 | 1.00 | 1.00 |
| DP5 | D5 | D6 | 200.00 | 43.00 | 50.09 | 46.00 | 48.90 | 44.81 | 1.19 | 1.19 | 1.00 | 1.00 |
| DP6 | D6 | D7 | 200.00 | 42.50 | 46.00 | 45.12 | 44.81 | 43.72 | 1.19 | 1.40 | 1.00 | 1.17 |
| DP7 | D7 | D8 | 250.00 | 18.90 | 45.12 | 45.00 | 43.72 | 42.61 | 1.40 | 2.39 | 1.17 | 2.10 |
| DP8 | D8 | D9 | 315.00 | 33.70 | 45.00 | 43.80 | 42.61 | 41.66 | 2.39 | 2.14 | 2.10 | 1.85 |
| DP9 | D9 | D10 | 315.00 | 37.00 | 43.80 | 42.80 | 41.66 | 40.32 | 2.14 | 2.48 | 1.85 | 2.19 |
| DP10 | D10 | D11 | 315.00 | 40.50 | 42.80 | 39.37 | 40.32 | 37.99 | 2.48 | 1.38 | 2.19 | 1.00 |
| DP11 | D11 | DE [*] | 400.00 | 10.70 | 39.37 | 39.13 | 37.99 | 37.75 | 1.38 | 1.38 | 1.00 | 1.00 |

Note: (1) upstream; (2) downstream; [*] existing manhole.

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
