# Peer review of "Influence of Green Roofs on the Design of a Public Stormwater Drainage System: A Case Study"

_sustainability, doi:10.3390/su15075762_

Round 1

Reviewer 1 Report

This article helps contribute the uptake of GR in cities and suggests way to integrate them with the design of stormwater drainage system.

Some improvements are summarised below: 

Results: Tables 3 to 9 can be moved to an appendix and replaced with tables/graphs/plot more effective to communicate results. Current result tables are quite overwhelming and do nto serve the purpose too well.

Results: cost-analysis should be better explained and integrated in the narrative.

Discussion: most the discussion section read like results. I'd recommend to investigate more the benefits for the public stormwater (i.e., how huge is the improvement as hinted in Line 253) 

Discussion: Lines 266-276 should be part of a separate sub-section when the financial analysis is highlighted.

Reviewer 2 Report

Dear authors,

The topic covered in the present paper is interesting and opportune. But before being taken into consideration for publication, I advise the following:

The results are only listed in the results section, which is presented insufficiently. The authors' instruction advises that this section should provide a concise description of the experimental findings, their interpretation, and any potentially problematic experimental conditions. I propose adding the missing data to the results (some of them must be brought here from the discussion section). In discussions, the findings must be interpreted from the standpoint of earlier studies.

Reviewer 3 Report

The topic is of great importance today, which justifies the publication of the article, although it is not very deep in the technical-scientific approach.

However, it is recommended to the authors a detailed review as they can improve the text. For example, on lines 187 and 188, the maximum and minimum velocities that regulations usually indicate for stormwater drainage pipes refers to design flows, and this clarification can be added.

The authors present and discuss a practical case to evaluate the benefits of using green roofs to reduce stormwater flows and floods in urban areas.

Being an increasingly important measure in some regions, due to climate change and increasing soil impermeabilization, it appears that there are still not many studies in this area. This article, although more technical than scientific, responds to that need.

As mentioned, it is yet another contribution to the dissemination of this constructive solution (green roofs). The fact that it is a very technical article, adds a new perspective, given that the few published articles are generally more theoretical.

As a pointedly technical article, the methodology is adequate. However, I believe that this is the main question that the journal must consider, that is, whether or not to accept this text as an article, bearing in mind the minor scientific contribution, on the one hand, and the scope of the journal, on the other. Personally, I value practical cases, but I admit that many readers of the journal do not have the same perspective and expect more theoretical contributions.

Figures can be considered adequate. However, Tables 3 to 9 can be considered a little excessive, as they contain some information that is not essential for conclusions (such as the Drag tension or Manhole base, for example). Can be reduced.

Reviewer 4 Report

Dear Authors, I have completed a review of the manuscript "Influence of green roofs on the design of a public stormwater drainage system: A case study". The article was written carefully, the content is well ordered and transparent. Unfortunately, a scientific element is lacking. The subject of using green roofs in cities is important and still topical, but it is not a novelty and has been presented in various aspects in many publications. The reviewed manuscript does not introduce anything new - it is obvious that the smallest diameter of the stormwater system pipes will be obtained in the scenario No. 3 with green roofs (this is due to the hydraulic calculation methodology), which also affects the costs. In its current form, manuscript is a report on design calculations rather than a scientific article.

However, I see the possibility to salvage this article by supplementing the analysis, e.g. you can estimate not only the costs of the stormwater system, but also roof costs; you can take into account not only investment costs, but also operating costs; you can take into account not only the financial aspect, but also environmental or social aspect, ... etc. This would give the basis to comparisons and choosing the optimal scenario.

Round 2

Reviewer 4 Report

Dear Authors, your response is very debatable. However, the additions made to the article, especially supplemented results and discussion, are sufficient to consider the article as having scientific elements. Thus, I recommend the article for publication in Sustainability.